# Marine-Steroid Derivative 5α-Androst-3β, 5α, 6β-triol Protects Retinal Ganglion Cells from Ischemia–Reperfusion Injury by Activating Nrf2 Pathway

**DOI:** 10.3390/md17050267

**Published:** 2019-05-05

**Authors:** Longxiang Sheng, Bingzheng Lu, Hui Chen, Yun Du, Chen Chen, Wei Cai, Yang Yang, Xuyan Tian, Zhaofeng Huang, Wei Chi, Suizhen Lin, Guangmei Yan, Wei Yin

**Affiliations:** 1Department of Pharmacology, Zhongshan School of Medicine, Sun Yat-sen University, Guangzhou 510080, China; shenglx@mail2.sysu.edu.cn (L.S.); lubzh3@mail2.sysu.edu.cn (B.L.); duyun3@mail2.sysu.edu.cn (Y.D.); chench376@mail2.sysu.edu.cn (C.C.); ygm@mail.sysu.edu.cn (G.Y.); 2State Key Laboratory of Ophthalmology, Zhongshan Ophthalmic Center, Sun Yat-sen University, Guangzhou 510060, China; 129huihui@163.com (H.C.); chiwei@mail.sysu.edu.cn (W.C.); 3Guangzhou Cellprotek Pharmaceutical, G Building F/4, 3 Lanyue Road, Science City, Guangzhou 510663, China; yangyang@cellprotek.com (Y.Y.); linsuizhen@cellprotek.com (S.L.); 4Institute of Human Virology, Zhongshan School of Medicine, Sun Yat-sen University, Guangzhou 510080, China; tianxy25@mail2.sysu.edu.cn (X.T.); hzhaof@mail.sysu.edu.cn (Z.H.); 5Department of Biochemistry, Zhongshan School of Medicine, Sun Yat-sen University, Guangzhou 510080, China; caiw28@mail2.sysu.edu.cn

**Keywords:** acute glaucoma, 5α-androst-3β, 5α, 6β-triol (TRIOL), NF-E2-related factor 2 (Nrf2), ischemic-reperfusion (I/R) injury

## Abstract

High intraocular pressure (IOP)-induced retinal ischemia leads to acute glaucoma, which is one of the leading causes of irreversible visual-field loss, characterized by loss of retinal ganglion cells (RGCs) and axonal injury in optic nerves (ONs). Oxidative stress and the inflammatory response play an important role in the ischemic injury of retinal and optic nerves. We focus on 5α-androst-3β, 5α, 6β-triol (TRIOL), a synthetic neuroactive derivative of natural marine steroids 24-methylene-cholest-3β, 5α, 6β, 19-tetrol and cholestane-3β, 5α, 6β-triol, which are two neuroactive polyhydroxysterols isolated from the soft coral *Nephthea brassica* and the gorgonian *Menella kanisa,* respectively. We previously demonstrated that TRIOL was a neuroprotective steroid with anti-inflammatory and antioxidative activities. However, the potential role of TRIOL on acute glaucoma and its underlying mechanisms remains unclear. Here, we report TRIOL as a promising neuroprotectant that can protect RGCs and their axons/dendrites from ischemic–reperfusion (I/R) injury in an acute intraocular hypertension (AIH) model. Intravitreal injection of TRIOL significantly alleviated the loss of RGCs and the damage of axons and dendrites in rats and mice with acute glaucoma. As NF-E2-related factor 2 (Nrf2) is one of the most critical regulators in oxidative and inflammatory injury, we further evaluated the effect of TRIOL on Nrf2 knockout mice, and the neuroprotective role of TRIOL on retinal ischemia was not observed in Nrf2 knockout mice, indicating that activation of Nrf2 is responsible for the neuroprotection of TRIOL. Further experiments demonstrated that TRIOL can activate and upregulate Nrf2, along with its downstream hemeoxygenase-1 (HO-1), by negative regulation of Kelch-like ECH (Enoyl-CoA Hydratase) associated Protein-1 (Keap1). In conclusion, our study shed new light on the neuroprotective therapy of retinal ischemia and proposed a promising marine drug candidate, TRIOL, for the therapeutics of acute glaucoma.

## 1. Introduction

Retinal ischemia has been well-recognized as a key pathological factor for various eye diseases, including acute glaucoma [1], diabetic retinopathy [2], and retinal arterial occlusion [3]. Among these, acute glaucoma is the most common and hazardous eye disorder, especially in East Asian ethnicities [1]. Ischemia–reperfusion (I/R) injuries of the retina in glaucoma are mainly caused by the rapid elevation and reduction of intraocular pressure (IOP), followed by apoptosis of retinal ganglion cells (RGCs), axonal damage, and subsequent vision loss. Although strategies of reducing IOP, such as filtering surgery and hypotensive agents, are widely used in clinical practice, it was also reported in the Early Manifest Glaucoma Trial that some patients with open-angle glaucoma cannot benefit from IOP-lowering therapy with no significant changes of the visual field [4,5]. The application of IOP-lowering agents, such as prostaglandin analogs and miotics, have also been reported with various local or systematic side effects [6,7]. Hence, the development of alternative strategies, such as neuroprotection-targeting oxidative and inflammatory responses, could support and supplement IOP-lowering therapy [8].

Recently, neuroactive marine steroids have been well-recognized as a new series of compounds showing neuroprotective effects on various central-nervous-system (CNS) diseases [9,10]. We previously investigated two natural marine steroids, 24-methylene-cholest-3β, 5α, 6β, 19-tetrol and cholestane-3β, 5α, 6β-triol, with neuroprotective effects. 24-methylene-cholest-3β, 5α, 6β, 19-tetrol is a tetrahydroxy sterol extracted from the soft coral *Nephthea brassica* collected in the South China Sea [11] that can protect cerebellar granule neurons from glutamate-induced excitotoxicity [12,13]. However, 24-methylene-cholest-3β, 5α, 6β, 19-tetrol was also reported to exhibit cytotoxic activities toward some cancer cells. Cholestane-3β,5α,6β-triol, on the other hand, is not only an endogenous metabolite of cholesterol, but also a natural marine steroid existing in various marine organisms, including soft coral *Sinularia sp* and gorgonian *Menella kanisa* [14,15]. It was proven in our previous works to protect rat brains from ischemic stroke in a middle-cerebral-artery occlusion (MCAO) model, and it protect rabbits from spinal cord I/R injury [16]. We further found that it can also suppress epileptic seizures by negative regulation of voltage-gated sodium channels [17]. In order to overcome the possible side effects of cholestane-3β,5α,6β-triol on vascular smooth muscle cell calcification [18], and the cytotoxic potential of 24-methylene-cholest-3β, 5α, 6β, 19-tetrol, their derivative, 5α-androst-3β,5α,6β-triol (TRIOL), lacking an isooctyl side chain, was designed and synthesized (Figure 1A). Retaining the basic polyhydroxy sterol structure of cholestane-3β, 5α, 6β-triol counts for the neuroactive activity of TRIOL, while the deletion of the hydrophobic isooctyl side chain improves the solubility of TRIOL in vehicles of hydroxypropyl-β-cyclodextrin (HP-β-CD) and may reduce side effects. Our previous in vitro studies demonstrated that TRIOL can serve as a potent neuroprotectant by protecting mitochondria with antioxidative and anti-inflammatory properties [19,20]. However, whether TRIOL can protect neurons from acute I/R injury in vivo, as well as its underlying mechanisms, remain largely unknown.

The Keap1/Nrf2/HO-1 system is one of the most important endogenous defenses against oxidative stresses, especially in retinal ischemic injury. Reperfusion of the retina after ischemia induces the overproduction of reactive oxygen species (ROS), reactive nitrogen species (RNS), and inflammatory responses in RGCs and microglia. Keap1 is a cysteine thiol-rich component of the Cullin–RING E3 ligase complex functioning as an ROS sensor [21]. In normal conditions, Keap1 inhibits Nrf2 activation by forming a ubiquitin E3 ligase complex with Cullin-3 and Nrf2, which ubiquitinates Nrf2 and enables the rapid and constant degradation of Nrf2. When cells were burdened with ROS insults, the exposed cysteine residue of Keap1 was modified by ROS, which reduces the activity of the Keap1–Cullin3 ubiquitin E3 ligase complex and stabilizes Nrf2 [22]. Then, the accumulated Nrf2 translocates into the nucleus, forms a heterodimer with one of the small Maf proteins, binds to antioxidant response elements (AREs), and activates the translation of various antioxidant genes, such as hemeoxyganse-1 (HO-1) [23]. Besides the critical role of Nrf2 in suppressing oxidant response, recent studies demonstrated that Nrf2 is also essential for inhibiting inflammatory responses. Nrf2 deficiency exacerbates inflammation in various disorders such as stroke [24], sepsis [25], and pneumonia [26]. Nrf2 also inhibits the microgliosis and release of inflammatory cytokines of microglia in experimental Parkinson’s Disease (PD) mice [27]. Given its antioxidative and anti-inflammatory properties, Nrf2 could alleviate the I/R injury of acute glaucoma, and might be a promising target for the development of therapeutic drugs.

Here, we report TRIOL as a novel marine neuroprotectant that can protect RGCs from acute I/R injury via inhibiting microglial activation by enhancing the Nrf2/HO-1 pathway in the acute intraocular hypertension (AIH) model, which may provide new insights and a promising candidate agent for the therapeutics of acute glaucoma.

## 2. Results

### 2.1. TRIOL Protects Retina from Ischemic–Reperfusion Injury in a Rat AIH Model

To investigate if TRIOL can serve as a neuroprotectant and alleviate the acute I/R injury of the retina in the acute phase of glaucoma, we first evaluated the neuroprotective effect of TRIOL in a rat AIH model. We chose intravitreal injection as a route of drug administration in order to increase local drug concentration in the retina. Limited by the volume capacity of rat vitreous cavity, the maximum injection volume of TRIOL (10 mg/mL) was 8 μL per eye. After AIH treatment on the right eyes, the right eyes of each group received an intravitreal administration of 40 or 80 μg TRIOL per eye, or 20% HP-β-CD as vehicle, respectively, while the left eyes served as normal control. As is shown in Figure 1B, the number of RGCs in the Ganglion Cell Layer (GCL) 20 μm adjacent to the optic disc were calculated to evaluate RGC injury. Contrasted with the dramatic loss of RGCs in the GCL of the AIH group, the injection of 80 μg TRIOL partially but significantly recovered the loss of RGCs (Figure 1C–E). Furthermore, as RGC axons and dendrites play the most important role in transferring information from the retina to the visual cortex in the brain, we also evaluated the integrity of RGC axons and dendrites by immunohistochemistry analysis of β-III-Tubulin (Tuj1). Compared with regular and dense Tuj1-labelled axons and dendrites in the control group, we found that AIH treatment induced a significant decrease and breaking of RGC axons and dendrites, while TRIOL dose-dependently attenuated the injury and kept the integrity of the axon networks. Consistent with the in vitro results from our previous report [19,20], we demonstrate here TRIOL as a neuroprotectant in a rat glaucoma model induced by AIH treatment.

### 2.2. Nrf2 Pathway Involved in Retinal Protection of TRIOL

Next, we sought to investigate whether the activation of Nrf2 is responsible for the neuroprotective effect of TRIOL using Nrf2 knockout mice. First, the retina protein samples were isolated from wild-type (wt) C57BL6J and Nrf2^−/−^ mice, and were used to assay the expression level of Nrf2 and its downstream antioxidant enzyme HO-1. As is shown in Figure 2A, Nrf2^−/−^ mice indeed did not express Nrf2 and HO-1. Then, we used the Nrf2^−/−^ mice to test whether Nrf2 knockout abolished the retinal protection of TRIOL in the AIH model. The hematoxylin–eosin (HE) staining of retina slices and counting RGC numbers in GCL 20 μm adjacent to ON, as described above (Figure 2B), were carried out to assay the neuroprotection effect of TRIOL. Consistent with the rat results, treatment of TRIOL significantly increased the number of RGCs after AIH insult in wt mice. However, the protective effects of TRIOL on RGC loss were not observed in Nrf2^−/−^ mice. Furthermore, we evaluated the injury of axons/dendrites in wt and Nrf2^−/−^ mice. The treatment of TRIOL significantly increased the fluorescence intensity of Tuj1 and repaired the network of axons/dendrites injured by AIH; these protective effects of TRIOL also disappeared in Nrf2^−/−^ mice (Figure 2E,F). Taken together, these results strongly indicate that the Nrf2 pathway contributed to the neuroprotective effect of TRIOL on retinal ischemia.

### 2.3. TRIOL Inhibits Inflammatory Activation of Microglia In Vitro and In Vivo

Recent studies reported that the activation of microglia in optic nerves can damage RGCs and axons in experimental glaucoma [28,29], and the activation of the Nrf2/HO-1 system can inhibit microglial activation [30,31]. The role of Nrf2 in inhibiting microglial activation and the possible anti-inflammatory property of TRIOL in our previous study [19,20] strongly suggested that the inhibition of microglial activation by activating the Nrf2/HO-1 pathway may be involved in the neuroprotection of TRIOL.

To evaluate the effect of TRIOL on microglial activation, we first checked the role of TRIOL on the activation of BV2 microglia induced by lipopolysaccharides (LPS). As is shown in Figure 3A, 100 ng/mL LPS treatment for 12 h resulted in typical activated amoeboid-like morphology of BV2 microglia, characterized by enlarged cell bodies and the formation of filopodia or lamellipodia, while the pretreatment of TRIOL (10 μM) significantly reduced the number of activated microglia with morphological changes (Figure 3B). In addition, TRIOL significantly reduced the mRNA expression of inflammatory cytokines TNF-α and IL-1β, and proinflammatory chemokines CXCL-10 and CCL-2, which are robustly stimulated by LPS (Figure 3C). To further confirm the inhibition of TRIOL on microglial activation in vitro, we then investigated the role of TRIOL on the expression of Iba-1, a well-established marker of microglial activation in the optic nerves of AIH mice by immunohistochemistry analysis. In Figure 3D and E, AIH retinal ischemia significantly increased the number of activated microglia with activated morphology, which was remarkably attenuated by the treatment of TRIOL. In conclusion, in vitro and in vivo results demonstrated the anti-inflammatory potential of TRIOL in microglia.

### 2.4. TRIOL Induces Nuclear Translocation of Nrf2 in Microglia

The nuclear translocation of Nrf2 is essential for its binding with ARE and activating the transcription of downstream antioxidant genes such as HO-1 [32]. To investigate whether TRIOL can inhibit microglial activation by activating the Nrf2/HO-1 pathway, we evaluated the protein levels and nuclear translocation of Nrf2 and HO-1 in vitro and in vivo. Using immunofluorescence imaging, we first analyzed the localization of Nrf2 in normoxia and an I/R condition with/without the treatment of TRIOL. In the normoxia condition, pretreatment of 10 μM TRIOL significantly induced the nuclear translocation of Nrf2 (Figure 4A,B) in BV2 microglia, indicating the activation of Nrf2, which was further confirmed by the increased fluorescence intensity of HO-1 (Figure 4A,C). To mimic I/R injury in vitro, BV2 microglia were treated with hypoxia stimuli (1% oxygen, 12 h) and subsequent reperfusion to normoxia (20% oxygen, 12 h). Immunofluorescence imaging of Nrf2 and HO-1 indicated that I/R treatment induced increased nuclear Nrf2 and HO-1 upregulation, which was further notably enhanced by the pretreatment of 10 μM TRIOL (Figure 4A–C). Next, we further analyzed the effect of TRIOL on microglial Nrf2 in the AIH model using wt C57BL6J mice. As is shown in Figure 4D, the treatment of TRIOL also increased the nuclear signal of Nrf2 in microglia, which was marked by Iba1. These observations indicated that TRIOL can activate the transcriptional activity of Nrf2 in microglia.

### 2.5. TRIOL Upregulates Expression of Nrf2 and HO-1 via Negative Regulation of Keap1

Besides analysis of TRIOL on the nuclear translocation of Nrf2, we also explored its regulation on Nrf2/HO-1 expression and possible mechanisms. First, we analyzed the effect of TRIOL on Nrf2 and HO-1 expression in the optic nerve by fluorescence imaging in the AIH model of wt and Nrf2^−/−^ mice. As shown in Figure 5A–D, Nrf2 and HO-1 were nearly undetectable in the optic nerve of Nrf2^−/−^ mice whether they were treated with AIH or TRIOL. In wt mice, AIH induced slight but also significant Nrf2 upregulation, while the treatment of TRIOL significantly enhanced the protein level of Nrf2 (Figure 5A,B). HO-1 expression was also sharply upregulated by TRIOL treatment (Figure 5C,D). As Keap1 was the key regulator on Nrf2 degradation via ubiquitin modification, we investigated if TRIOL could modulate Keap1 and promote the activation of Nrf2 in BV2 microglia. As is shown in Figure 5E,F, pretreatment of 10 μM TRIOL significantly inhibited the protein expression of Keap1 both under a normoxia condition or I/R treatment, accompanied by the notable increase of Nrf2/HO-1, which suggested that TRIOL upregulated Nrf2 by negative regulation of Keap1-mediated degradation.

## 3. Discussion

As the most frequent global cause of irreversible blindness, acute glaucoma remains an awkward hazard, and the exploration of safe and effective therapeutic strategies, especially neuroprotectants, is of great importance. Currently, the most widely accepted clinical treatment for acute glaucomatous optic neuropathy is to lower IOP, but most of these kinds of agents have side effects. As the first-line clinic drugs used to lower IOP, prostaglandin analogs such as latanoprost have the local side effects of enophthalmos and periorbitopathy [6,33,34], while β adrenergic blockers such as betaxolol have the risk of myocardial infarction, arrhythmias, and obstructive bronchial problems [35,36]. The treatment of miotics represented by pilocarpine often results in pupillary constriction, limiting their wide application in dark conditions [7]. Moreover, results from the Early Manifest Glaucoma Trial elucidated that reducing IOP alone cannot completely alleviate the progression of open-angle glaucoma [5]. These reports highlight the urgent need for novel therapeutic strategies with, for example, the combination of neuroprotectants and IOP-lowering agents. Currently, the shortage of safe and effective neuroprotectants limits their application in clinical use for the treatment of acute and chronic CNS disorders such as ischemic stroke, traumatic brain injury, and acute glaucoma. Current results, along with our previous studies of TRIOL, have preliminarily proven its potent neuroprotective effect against mitochondrial oxidative stress, microglial inflammatory activation, and neuronal excitotoxicity injury, suggesting TRIOL is a promising therapeutic candidate for acute glaucoma. However, glaucoma-associated cell injury is not only limited in RGCs, but also in photoreceptor cells [37], amacrine cells [38], and neurons in the visual cortex [39], which functions in different steps of visual formation. Further studies are required to investigate the potential role of TRIOL on the survival and function of these cells.

In the present study, we demonstrated that the activation of the Nrf2/HO-1 pathway contributes to the neuroprotective effect of TRIOL. These observations imply that Nrf2 is an important pharmacological target for acute glaucoma. In Figure 4 and Figure 5, we also found that AIH injury causes a slight nuclear translocation of Nrf2 and the upregulation of Nrf2/HO-1, which may represent an endogenous protective reaction as a response to oxidative and inflammatory insults. Retinal injury happens when the endogenous activation of Nrf2/HO-1 cannot deal with I/R insults, while TRIOL activated the Nrf2/HO-1 endogenous protective pathway by suppressing Keap1. However, more studies are needed to investigate the details of how TRIOL regulates Keap1.

Consistent with our observations of TRIOL, there are indeed several other compounds that have been reported to alleviate retinal ischemia by inducing Nrf2/HO-1 in experimental glaucoma, such as resveratrol [40,41] and flavonoid [42]. However, the effects of these agents on the injury of RGC axons have mostly been neglected or ignored. Our findings in Figure 1F and Figure 2E demonstrated the protection of TRIOL on RGC neuronal axons and dendrites, indicating the potential function of TRIOL on regulating the axonal integrity or the resprouting of axons after I/R injury, to which more attention should be paid in the following studies. More importantly, as the retinofugal and corticofugal axons play key roles in signal transmission from the retina to the visual cortex, our results here implied that the activation of the Nrf2/HO-1 pathway may contribute to maintaining the integrity of retinal axons and keeping visual function. It is noticeable that HO-1, which is transcriptionally activated by Nrf2, is a rate-limiting hemeoxygenase functioning by hydrolyzing heme into ferritin, biliverdin, and carbon monoxide [23]. The antioxidative effect of TRIOL by upregulating HO-1 may be attributed to the production of catalysate products, among which biliverdin is a strong reductive that can scavenge free radicals and inhibit lipid peroxidation [43]. The heme-scavenging system represented by HO-1 provides endogenous protection against secondary oxidative injury in glaucoma. Collectively, our findings here indicate the vital role of the Nrf2/HO-1 pathway in retinal protection, suggesting Nrf2 is a possible drug target for retinal ischemia.

## 4. Materials and Methods

### 4.1. Animals and Ethics Statements

All animal experiments were operated in accordance with the standards of the Animal Research: Reporting of In Vivo Experiments (ARRIVE) guidelines [44], and the experiment projects were approved by the Ethics Committee of Zhongshan Ophthalmic Center of Sun Yat-sen University (approval number 2018-111). Male Sprague Dawley (SD) rats were purchased from the Guangdong Medical Laboratory Animal Center (Guangzhou, China), while wild-type C57BL6J mice were purchased from the Nanjing Biomedical Research Institute (NBRI) of Nanjing University. Nrf2^−/−^ knockout mice were kind gifts from Professor Huang in the Zhongshan School of Medicine of Sun Yat-sen University. All animals were housed under environmentally controlled conditions, and efforts were made to minimize the number of experiment animals, to ensure their welfare, and to alleviate their suffering. Sixty SD rats, 40 wild-type C57BL6J mice, and 40 Nrf2^−/−^ knockout mice were used in this research. The operation and analysis of all animal experiments in this study were performed blindly and randomly.

### 4.2. AIH Model

The AIH model was operated as previously described with slight modification [45]. After systemic anesthesia with intraperitoneal injection of sodium pentobarbital (50 mg/kg) combined with corneal tropical anesthesia with 0.5% tetracaine hydrochloride, 1% tropicamide was dropped in the unilateral (right) eyes to dilate the pupils. The anterior chamber of the right eyes was cannulated with 30 gauge infusion needles that were connected to an elevated saline reservoir (0.9% NaCl). Intraocular pressure of the right eyes was raised by 130 mmHg and maintained for 60 min, with the left eyes serving as normal controls. Successful ischemia of the retina was confirmed by the whitening of the iris due to the collapse of the central arteries of the retina. Sixty minutes after retinal ischemia, TRIOL (10 mg/mL, Guangzhou Cellprotek Pharmaceutical Co. Ltd., Guangzhou, China) or vehicle (20% HP-β-CD) was intravitreally injected to the right eyes for 1 min using a microsyringe with a 30 gauge needle. Then, the right eyes were treated with tobramycin and dexamethasone sodium phosphate eye ointment. Individuals were excluded from analysis only upon death, model failure, or retinal hemorrhage happening during intravitreal injection. Forty-eight hours after AIH surgery, the rats/mice were deeply anesthetized with sodium pentobarbital (100 mg/kg, i.p.); then, retinas with optic nerves were isolated and fixed. Tissue samples were first fixed in an optimized FFA fixative solution (glacial acetic acid:formaldehyde solution:saline:75% ethanol = 1:2:7:10 (V/V)) for 6 h, followed by 4% paraformaldehyde for 42 h. Subsequent treatment of tissue samples for pathologic examination is described below.

### 4.3. Cell Culture and Hypoxia Treatment

BV2 microglia (China Center for Type Culture Collection, CCTCC, Wuhan, China) cells were cultured in Dulbecco’s Modified Eagle Medium (Gibco, Thermofisher Scientific, USA) with 10% FBS (Gibco, Thermofisher Scientific, Rockford, IL, USA). Upon 80%–90% confluency, BV2 cells were trypsinized using 0.25% trypsin (Gibco, Thermofisher Scientific, Rockford, Illinois, USA) and seeded on different dishes or confocal coverglass chambers according to the experiment design. Cell cryopreservation and recovery were operated in routine methods. The hypoxia treatment of BV2 microglia was carried out using an oxygen-controlled hypoxic working station (HypoxyLabTM, Oxford Optronix, UK). Briefly, cultured BV2 microglia were placed in a sterile chamber with 1% O_2_, 5% CO_2_, 90 ± 1% humidity (RH), and 37.0 ± 0.5 °C temperature. The fixation or protein extraction of hypoxic cells was operated in the hypoxic condition of the chamber.

### 4.4. Immunofluorescence Confocal Imaging

All cultured cells for immunofluorescence imaging were seeded on confocal chamber slides (Lab-Tek, Thermofisher Scientific, Rockford, IL, USA). After pretreatment of drug or hypoxia stimulus, cells were immediately fixed in 4% paraformaldehyde for 20 min at room temperature, followed by washing 3 times in Phosphate Buffered Saline (PBS). Samples were then permeabilized in 0.2% Triton/PBS for 15 min, followed by the incubation of primary antibodies against Nrf2 (Cell Signaling, 14596, Danvers, MA, USA), Iba1 (Wako, 017-19741, Osaka, Japan), or HO-1 (Abcam, ab13248, Cambridge, MA, USA) overnight at 4 °C in an antibody Diluent with Background Reducing Components (DAKO, S3022, Santa Clara, CA, USA). The second-day cells were washed and incubated with intended fluorescence-conjugated secondary antibodies (Molecular Probes, Thermofisher Scientific, Rockford, IL, USA) for 1 h. Then, cells were washed and stained with Hoechst33342 (5 μg/mL, Sigma Aldrich, St. Louis, MO, USA). For the procedure of tissue sections, retina and nerve samples were routinely isolated and fixed. Then, samples were embedded in paraffin and cut into 5 μm thick slices. Slices were deparaffinized and hydrated, followed by antigen retrieval using microwaves in EDTA retrieval solution (BOSTER Biotech, Wuhan, China) and the same procedures as described above in cultured cells. Confocal images were captured using a Nikon A1 Spectral Confocal Microscope (Nikon, Tokyo, Japan). Images of different groups were captured with the same parameters of pinhole (1.2 AU), resolution, laser power, and HV voltage for each channel to ensure the accuracy of intensity value. Images for cultured cells were captured with a magnification of 600×, while retina-tissue slices with a magnification of 200×.

### 4.5. Immunohistochemistry and Hematoxylin–Eosin Staining

Retina and nerve tissue samples intended for immunohistochemistry (IHC) were routinely treated as described above, and IHC staining was operated according to the manufacturer’s instructions of the IHC staining kit (Abcam, ab80436, Cambridge, MA, USA). Briefly, after the gradients of deparaffinization and hydration, slices were placed with antigen retrieval for 30 min, and in 3% hydrogen peroxide for 15 min. Cooling to room temperature, samples were incubated with indicated primary antibodies against Iba-1 (Wako, 017-19741, Osaka, Japan) overnight at 4 °C and HRP-conjugates for 30 min. Washing with PBS 3 times, slices were stained with a Diaminobenzidine (DAB) substrate–chromogen mixture and hematoxylin in turn. HE staining of retinal tissue was routinely operated according to the manufacturer’s instructions of the staining kit (Nanjing Jiancheng Technology Ltd., Nanjing, China).

The IHC and HE images of the retina and nerve slices were captured using an automatic digital slide scanning microscope (AxioScan.Z1, Zeiss, Germany) and an inverted microscope (Eclipse Ti-U, Nikon, Japan). Image analysis, including the counting of RGC numbers, was carried out blindly using Image-Pro Plus software (version 6.0, Media Cybernetics, Inc., Silver Springs, MD, USA).

### 4.6. Western Blotting

Cells were washed with PBS and lysed with M-PER™ Mammalian Protein Extraction Reagent (Thermofisher Scientific, Rockford, IL, USA) supplemented with 1× protease inhibitor cocktail (Merck Millipore, Cat No. 539131, San Diego, CA, USA), while retina-tissue samples were homogenized and lysed with a RIPA lysis buffer (Beyotime, Shanghai, China) with protease inhibitor cocktail, followed by centrifugation and supernatant collection. A BCA protein assay kit (Thermofisher Scientific, Rockford, IL, USA) was used to detect the concentration of total proteins. About 10 μg of total proteins was loaded and separated by 10% SDS-PAGE gels, followed by western blotting using antibodies against Nrf2 (Abcam, ab137550, Cambridge, MA, USA), HO-1 (Abcam, ab13248, Cambridge, MA, USA), Keap-1 (Cell Signaling, 8047s, Danvers, MA, USA), and α-Tubulin (Cell Signaling, 2125s, Danvers, MA, USA). After the respective incubation of the secondary antibodies, blottings were exposed with immobilon western chemilum HRP substrate (Merck Millipore, San Diego, CA, USA) and captured using a Bio-Rad ChemiDoc XRS+ imaging system (Bio-Rad Laboratories, Hercules, CA, USA).

### 4.7. Real-Time Reverse Transcript PCR

Total RNA was extracted from the cultured BV2 microglia under different treatment conditions using a TRIzol^®^ reagent (Thermofisher Scientific, Rockford, IL, USA, 15596) according to the manufacturer’s standard procedure, followed by reverse transcription with RevertAid Reverse Transcriptase (Thermofisher Scientific, Rockford, IL, USA, EP0442). SuperReal qPCR PreMix (SYBR Green) (Tiangen, FP202, Beijing, China) was prepared according to the manufacturer’s instructions, and real-time qPCR was conducted using a 7500 fast real-time qPCR system (Applied Biosystem, Foster City, CA, USA). PCR thermal cycling was initiated by a holding stage at 95 °C for 15 min, followed by a cycling stage (40 cycles) of 95 °C for 10 s, 62 °C for 30 s, 72 °C for 30 s, and a melt-curve stage. Specific primer pairs of inflammatory cytokines and chemokines used in this study were synthesized by Invitrogen (Thermofisher Scientific, Rockford, IL, USA), and their sequences are listed below. Relative gene expression was analyzed using a classical ∆∆Ct method previously reported by Schmittgen TD and Livak KJ [46].

β-actin (NM_007393.5) forward: 5’-AGATCAAGATCATTGCTCCTCCT-3’; reverse: 5’-ACGCAGCTCAGTAACAGTCC-3’;

TNF-α (NM_001278601.1) forward: 5’- AGGCACTCCCCCAAAAGATG-3’; reverse: 5’- CCACTTGGTGGTTTGTGAGTG-3’;

IL-1β (NM_008361.4) forward: 5’- TGCCACCTTTTGACAGTGATG-3’; reverse: 5’- AAGGTCCACGGGAAAGACAC-3’;

CXCL-10 (NM_021274.2) forward: 5’- CCACGTGTTGAGATCATTGCC-3’; reverse: 5’- GAGGCTCTCTGCTGTCCATC-3’;

CCL-2 (NM_011333.3) forward: 5’- GCTGTAGTTTTTGTCACCAAGC-3’; reverse: 5’- AAGGCATCACAGTCCGAGTC-3’.

### 4.8. Statistical Analysis

All data are presented as mean ± SD and were analyzed with Graphpad Prism 6 software (Graphpad Software. Inc., San Diego, CA, USA). One-way ANOVA was performed to analyze the data from multiple groups, followed by multiple comparisons using Dunnett’s post hoc test. *p* < 0.05 was considered statistically significant.

## 5. Conclusions

In this study, we reported for the first time that 5α-androst-3β, 5α, 6β-triol, a novel marine neuroactive steroid, can protect and alleviate the injury of RGCs and their axons/dendrites in an experimental acute glaucoma model. Further exploration identified that TRIOL can protect the retina by inhibiting microglial activation via enhancing the Nrf2/HO-1 pathway. These results may provide new insights into understanding of the treatment, and provide an ideal candidate compound for the drug discovery of acute glaucoma.

## Figures and Tables

**Figure 1 marinedrugs-17-00267-f001:**
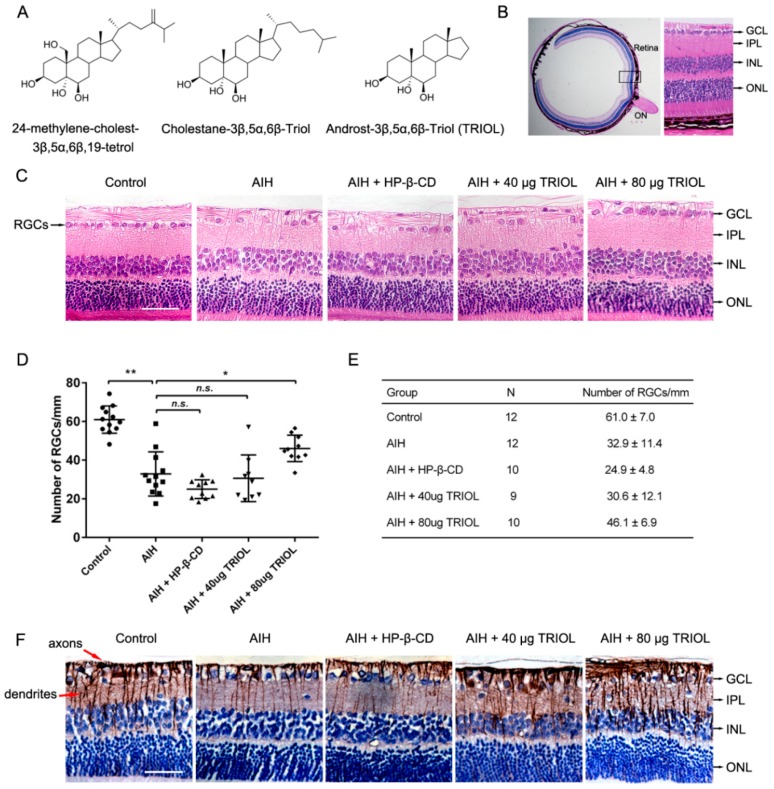
Derivative 5α-androst-3β, 5α, 6β-triol (TRIOL) protects the retina from ischemic–reperfusion injury in a rat acute intraocular hypertension (AIH) model. (**A**) Chemical structure of 24-methylene-cholest-3β, 5α, 6β, 19-tetrol, cholestane-3β, 5α, 6β-triol, and TRIOL. (**B**) Schematic structural diagram of rat retina used for the analysis of retinal ganglion cells (RGCs). Black box indicates area of retina (20 μm adjacent to optic disc) used to analyze RGC survival (Optic Nerve (ON), Ganglion Cell Layer (GCL), Inner Plexiform Layer (IPL), Inner Nuclear Layer (INL), Outer Nuclear Layer (ONL)); scale bar, 400 μm. (**C**) Representative image of hematoxylin–eosin (HE) retina staining in different groups for analysis of RGCs in GCL. RGCs in GCL are marked with black arrows in the left panel; cell layers marked with black arrows in the right panel; scale bar, 50 μm. (**D**,**E**) Quantification of RGC numbers in GCL per mm in retina. Average RGC numbers per mm were normalized according to GCL length in each individual. N and average number are shown in **E**. Data are presented with mean ± SD. Statistical analysis of RGC numbers was performed using one-way ANOVA followed by Dunnett’s post hoc analysis. *N.s.*, no significance; *, *p* < 0.05; **, *p* < 0.01. (**F**) Representative images of immunohistochemistry (IHC) staining of Tuj1, marker of RGC axons and dendrites. Axons and dendrites are marked with red arrows in the left panel; cell layers marked with black arrows in the right panel; scale bar, 50 μm.

**Figure 2 marinedrugs-17-00267-f002:**
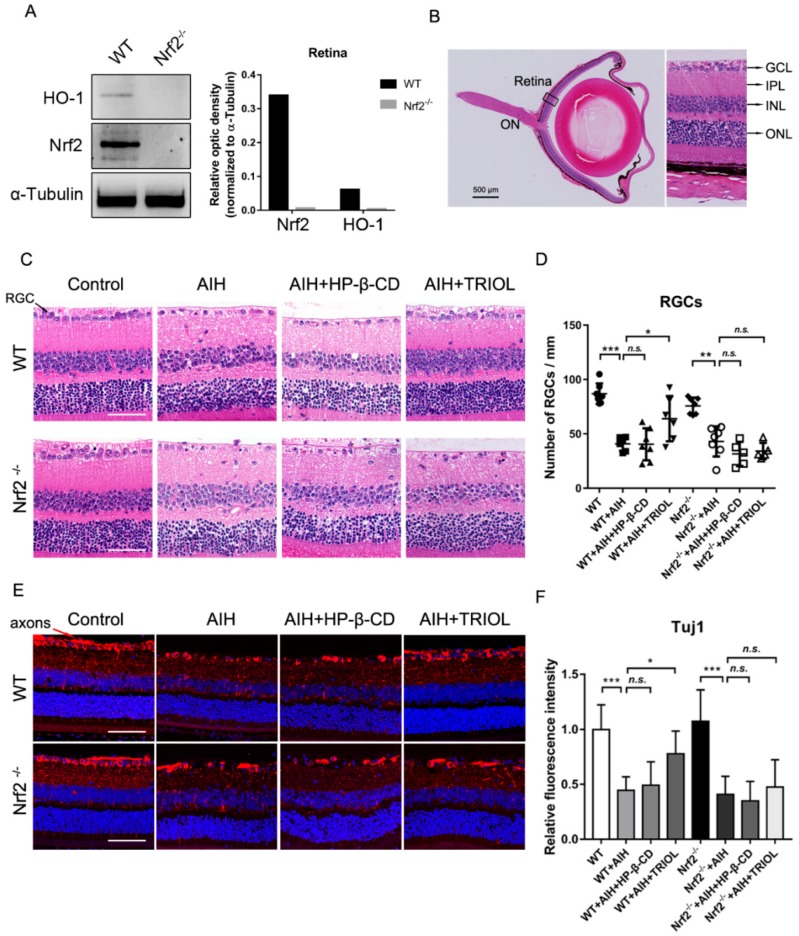
Nrf2 pathway is involved in the retinal protection of TRIOL. (**A**) Expression of Nrf2 and its downstream HO-1 were examined by western blot in the lysate of isolated retina from wild-type (wt) and Nrf2^−/−^ mice, shown in the left panel. Quantification of the optic density of Nrf2 and HO-1 in wt and Nrf2^−/−^ mice is shown in the right panel. (**B**) Schematic structural diagram of mouse retina used for RGC analysis. Black box, retina area to be analyzed; scale bar, 500 μm. (**C**) Representative images of HE retina staining for evaluation of RGC injury in different groups. RGCs in GCL are marked with black arrows in the left panel; scale bar, 50 μm. (**D**) Quantification of RGC numbers in the GCL of different groups in **C**. Calculation of average RGC numbers was described above. N = 5–7 mice for each group. (**E**) Representative confocal immunofluorescence imaging of RGC axons and dendrites stained by Tuj1 (red). RGC axons are marked with red arrows in the left panel. Nucleus was stained with Hoechst 33342 (blue). Magnification, 200×. (**F**) Quantification of relative fluorescence intensity of Tuj1 was measured using Nikon NIS-Elements AR software. Statistical analysis of RGC numbers in **D** and fluorescence intensity in **F** was performed using one-way ANOVA, followed by Dunnett’s post hoc analysis. *N.s.*, no significance; *, *p* < 0.05; **, *p* < 0.01; ***, *p* < 0.001.

**Figure 3 marinedrugs-17-00267-f003:**
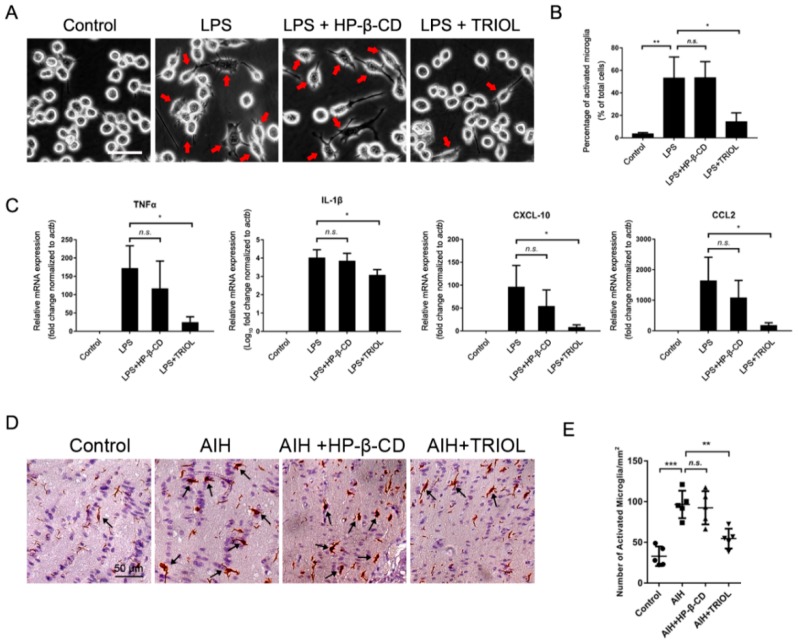
TRIOL inhibits the inflammatory activation of microglia in vitro and in vivo. (**A**) Representative phase-contrast morphological images of BV2 microglia treated with or without lipopolysaccharides (LPS) or TRIOL. BV2 microglia were pretreated with 10 μM TRIOL or vehicle (20% HP-β-CD) 30 min before the treatment of 100 ng/mL LPS. Phase-contrast images were captured 12 h after LPS treatment to observe the activated morphology of microglia. Red arrows indicate activated microglia with typical amoeboid-like morphology; scale bar, 20 μm. (**B**) Number of activated microglia in independent regions of interest (ROI) of different groups were blindly calculated using Image Pro Plus software. (**C**) MRNA expression of inflammatory cytokines TNF-α and IL-1β, and inflammatory chemokines CXCL-10 and CCL2, was detected using real-time polymerase chain reaction (PCR). N = 3 independent experiments for each group. (**D**) Representative images of IHC staining by Iba-1 in the optic nerves of wt mice treated with/without TRIOL in the AIH model. Black arrowheads, activated microglia with activated morphology characterized by enlarged cell bodies, more filopodia, and dense Iba1 staining. Quantification of activated microglia shown in **E**. Statistical analysis of cell numbers in **B** and **E**, and relative mRNA expression in **C**, was performed using one-way ANOVA, followed by Dunnett’s post hoc test. Data are shown in mean ± SD. *N.s.*, no significance; *, *p* < 0.05; **, *p* < 0.01; ***, *p* < 0.001.

**Figure 4 marinedrugs-17-00267-f004:**
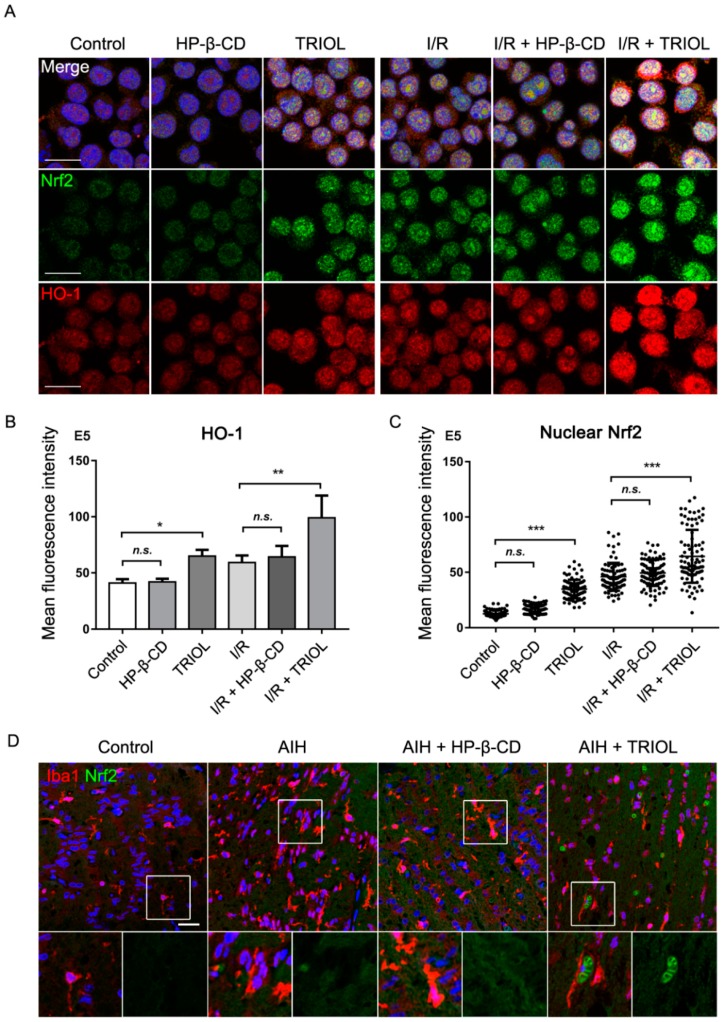
TRIOL induces the nuclear translocation of Nrf2 in microglia. BV2 microglia were pretreated with/without 10 μM TRIOL or vehicle (HP-β-CD) 1 h before treatment of 1% O_2_ hypoxia stimuli for 12 h, followed by reperfusion to normoxia (20% O_2_) for 12 h. (**A**) Expression and nuclear localization of Nrf2 and HO-1 were measured using immunofluorescence staining with antibodies against Nrf2 (green) and HO-1 (red) in BV2 microglia. Nucleus was stained with Hoechst 33342 (blue); scale bar, 20 μm. (**B**) Relative HO-1 fluorescence intensity was measured by NIS-Elements AR. N = 4 independent ROIs for each group. (**C**) Nrf2 intensity accumulated in a single nucleus was also measured to evaluate the activation of Nrf2; N = 90 cells for each group. (**D**) Representative confocal imaging of Iba1 (red) and Nrf2 (green) in the optic nerves of wt mice in different groups. Nucleus was stained with Hoechst 33342 (blue). Fields in white boxes were enlarged in the panel below; scale bar, 25 μm. Statistical analysis of fluorescence intensity in **B** and **C** was performed using one-way ANOVA, followed by Dunnett’s post hoc test. *N.s.*, no significance; *, *p* < 0.05; **, *p* < 0.01; ***, *p* < 0.001.

**Figure 5 marinedrugs-17-00267-f005:**
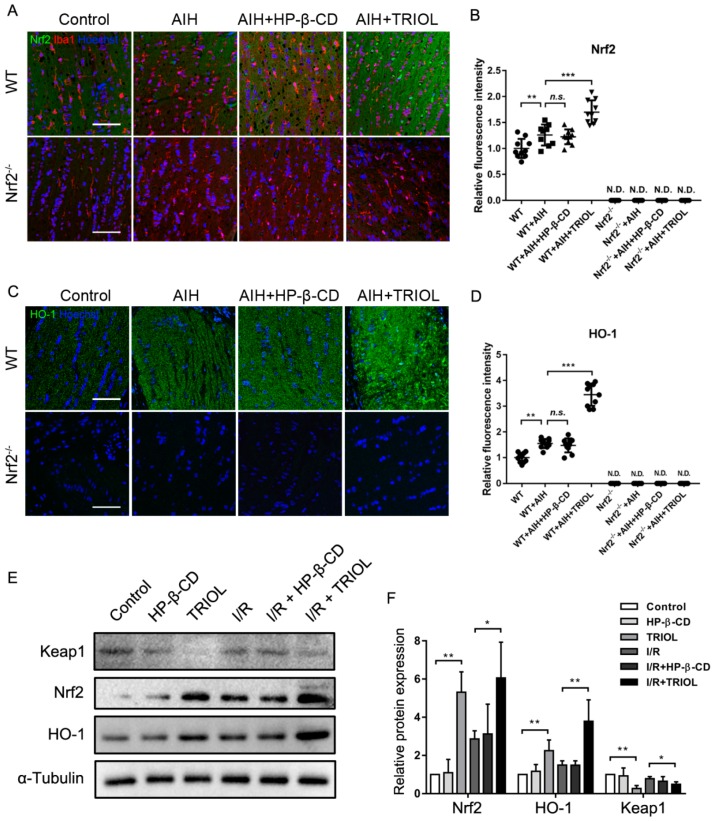
TRIOL upregulates expression of Nrf2 and HO-1 via negative regulation of Keap1. (**A**) Representative immunofluorescence images of Nrf2 (green) and Iba1 (red) in the optic nerves of different groups. Nucleus was stained with Hoechst 33342 (blue); scale bar, 100 μm. Quantification of relative Nrf2 fluorescence intensity shown in (**B**); representative confocal images and relative fluorescence intensity of HO-1 shown in (**C**,**D**), respectively. (**E**) Protein expression levels of Nrf2, HO-1, and Keap1 influenced by TRIOL and I/R stimuli were measured by western blot; representative images shown in **E**. Statistical analysis of relative protein expression measured by the gray-scale of blots shown in **F**. Statistical analysis of fluorescence intensity in **D** and **E** and relative protein expression in **F** were performed using one-way ANOVA, followed by Dunnett’s post hoc analysis. N.D., not detected; *n.s*., no significance; *, *p* < 0.05; **, *p* < 0.01; ***, *p* < 0.001.

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
