# Peer review of "Marine-Steroid Derivative 5α-Androst-3β, 5α, 6β-triol Protects Retinal Ganglion Cells from Ischemia–Reperfusion Injury by Activating Nrf2 Pathway"

_marinedrugs, 2019, doi:10.3390/md17050267_

Reviewer 1 Report

The authors present the data on the protective activity of a synthetic derivative i.e. TRIOL against oxidative stress and inflammation. Once they have established that its protective role against loss of retina ganglion cells they move on to provide more mechanistic evidence by showing genetic data in nrf2-/- mice that TRIOL mainly acts through NRF2. Overall the work is well structured and it may be of interest to wider readership. 

My major concern mainly reflects the novelty of the finings. In brief, TRIOL has been role for its antioxidant role. NRF2 has also been shown to have a therapeutic role in glaucoma. Thus, the authors need to further strengthen the novel findings of this work in relation to others.  

Minor points:

Figure 1C and F or 2C and E and elsewhere: arrows could be used to indicate the relevant layers or where the reader should focus on.

Figure 4A and D: it is not entirely clear that the authors have used the same confocal intensity parameters. If so, this should be stated.

Author Response

Point 1: My major concern mainly reflects the novelty of the findings. In brief, TRIOL has been role for its antioxidant role. NRF2 has also been shown to have a therapeutic role in glaucoma. Thus, the authors need to further strengthen the novel findings of this work in relation to others.

Response 1: We truly appreciated your insight advices here, which are very helpful for improving the quality of this MS and our research in the future. As is shown in the last paragraph of Introduction section and Abstract section (Line 41-43) of MS, we further strengthen the novelty as listed below:

1. Firstly, the pharmaceutical therapy for acute glaucoma has been challenged with various factors including the lack of safe and effective neuroprotectants. As we have mentioned in the Introduction (Line 53-58), although IOP lowering strategies are widely used clinically, they are also challenged by various local and systematic side effects and the unbeneficial outcomes of partial open-angle glaucoma patients which were reported in the EMGT trial [1, 2]. As an alternative complement strategy independent of IOP lowering strategies, neuroprotection has been considered of great importance to maximizing the recovery of RGCs and minimizing secondary insults [3]. However, several developmental attempts of neuroprotectants, such as NMDAR antagonist Memantine, failed to reduce visual field loss in phase III clinical trial (ClinicalTrials.gov. NCT00168350) [4, 5]. Currently, there are no effective neuroprotectants approved for the clinical application of glaucoma. In this study, we characterized and provided TRIOL as a promising lead compound and candidate drug for the development of neuroprotectants for glaucoma.

2. Secondly, as you have mentioned in the comments, we have previously reported the antioxidant role of TRIOL on glutamate-induced injury in primary rat cortical neurons with unknown mechanisms in vitro [6]. However, in the current MS, we further characterized the anti-inflammatory and antioxidant role of TRIOL in microglia in an experimental glaucoma model in vivo. Furthermore, by using Nrf2-/- knockout mice, we demonstrated for the first time that TRIOL can exert its neuroprotective effect by activating Nrf2/HO-1 pathway, which further suggested the vital role of Nrf2 as a target for the drug development of glaucoma.

Point 2: Figure 1C and F or 2C and E and elsewhere: arrows could be used to indicate the relevant layers or where the reader should focus on.

Response 2: Thank you for your vital advices on our data presentation.

1. We marked the RGCs, the axons/dendrites of RGCs and different cell layers in retina with conspicuous arrows in re-edited Figure 1.C/1.F and Figure 2.C/2.E. We hope this will make it more convenient for the readers to focus on the key information.

2. Relevant descriptions about the modifications mentioned above were added in the figure legends respectively. For Figure 1, the statement “RGCs in GCL were marked with black arrows in the left panel, with cell layers marked with black arrows in the right panel.” and “The axons and dendrites were marked with red arrows in the left panel, with cells layers marked with black arrows in the right panel.” were added in Line 139-140 and Line 146-147 respectively. For Figure 2, the statement “RGCs in GCL were marked with black arrows in the left panel.” and “Axons of RGCs were marked with red arrows in the left panel.” were added in Line 173 and Line 177 of the updated MS respectively.

Point 3: Figure 4A and D: it is not entirely clear that the authors have used the same confocal intensity parameters. If so, this should be stated.

Response 3: We are very sorry for the insufficient description of our methods regarding to confocal immunofluorescence imaging. The statement “Images of different groups were captured with the same parameters of pinhole (1.2AU), resolution, laser power and HV voltage for each channel to make sure the accuracy of intensity value. Images for cultured cells were captured with the magnification of 600×, while the retina tissue slices with the magnification of 200×. ” were added in 4.4 section of Materials and methods (Line 386-390) of the updated MS.

References:

1.    Heijl, A.; Leske, M. C.; Bengtsson, B.; Hyman, L.; Bengtsson, B.; Hussein, M., Reduction of intraocular pressure and glaucoma progression: results from the Early Manifest Glaucoma Trial. Archives of ophthalmology (Chicago, Ill. : 1960) 2002, 120, (10), 1268-79.

2.    Bengtsson, B.; Heijl, A., Lack of Visual Field Improvement After Initiation of Intraocular Pressure Reducing Treatment in the Early Manifest Glaucoma Trial. Investigative ophthalmology & visual science 2016, 57, (13), 5611-5615.

3.    Pardue, M. T.; Allen, R. S., Neuroprotective strategies for retinal disease. Progress in retinal and eye research 2018, 65, 50-76.

4.    Greenfield, D. S.; Girkin, C.; Kwon, Y. H., Memantine and progressive glaucoma. Journal of glaucoma 2005, 14, (1), 84-6.

5.    Zhang, K.; Zhang, L.; Weinreb, R. N., Ophthalmic drug discovery: novel targets and mechanisms for retinal diseases and glaucoma. Nature reviews. Drug discovery 2012, 11, (7), 541-59.

6.    Chen, J.; Leng, T.; Chen, W.; Yan, M.; Yin, W.; Huang, Y.; Lin, S.; Duan, D.; Lin, J.; Wu, G.; Zhang, J.; Yan, G., A synthetic steroid 5alpha-androst-3beta,5,6beta-triol blocks hypoxia/reoxygenation-induced neuronal injuries via protection of mitochondrial function. Steroids 2013, 78, (10), 996-1002.

Reviewer 2 Report

With the manuscript marinedrugs-489907, the authors report the potential therapeutic utility of 5α-androst-3β,5α, 6β-triol in the treatment of acute glaucoma, relevant data on the interference with Nrf2/HO-1 pathway being provided. While the results are relevant, it is my opinion that the content of the manuscript is not aligned with the scope of Marine Drugs.

It is unreasonable to classify 5α-androst-3β,5α, 6β-triol as “marine-derived” insofar as it is structurally similar to a series of ubiquitous steroids. Also, the steroid components referred by the authors in the current manuscript are common and not at all restrictively inspired in secondary metabolites isolated from marine sources. Additionally, while the authors mention the synthesis of analogues derived the soft-coral derived 24-methylene-cholest-3β,5α,6β,19-tetrol (Lines 63-65 and 68-70), including the compound under study, no data regarding their synthesis is provided.

In addition to considering that the content of the manuscript is not aligned with the scope of this journal, there are some additional comments that should be considered by the authors.

Although generally well written, English language editing is mandatory, particularly the sections of Introduction and Discussion. Examples include:

Line 50: “…followed with the by apoptosis of RGCs…”.

Line 55: Correct “moitics” to “miotics”.

Lines 61-62: “…natural product extracted from the soft coral Nephthea brassica collected in South China Sea…”.

Lines 68-70: “In order to overcome possible side effects of cholestane-3β,5α,6β-triol on endothelial cell calcification, its the analogue of 5α-androst-3β,5α, 6β-triol (TRIOL) without lacking the isooctyl side chain were was synthesized.”

Concerning the experimental data, the authors are requested to clarify the concentrations (80 μg ?) used on the in vivo testing in AIH rat models (Lines 103, 104 and 107). In order to validate the results from western blot analysis, i.e. the interference with the expression Nrf2 and HO-1 (section 2.2), it is also mandatory to include the results concerning the quantification of the relative optical density of the bands.

Author Response

Point 1: It is unreasonable to classify 5α-androst-3β,5α,6β-triol as “marine-derived” insofar as it is structurally similar to a series of ubiquitous steroids. Also, the steroid components referred by the authors in the current manuscript are common and not at all restrictively inspired in secondary metabolites isolated from marine sources. Additionally, while the authors mention the synthesis of analogues derived the soft-coral derived 24-methylene-cholest-3β,5α,6β,19-tetrol (Lines 63-65 and 68-70), including the compound under study, no data regarding their synthesis is provided.

Response 1: Thank you for your vital concerns about the structure, sources and synthesis of TRIOL.

1.      Indeed, 5α-androst-3β,5α,6β-triol (TRIOL) possesses a structure similar to a series of polyhydroxysterols. However, our previous studies [1] and results in this MS demonstrated that TRIOL is a neuroactive derivative which was designed and synthesized based on the structures of natural marine steroids 24-methylene-cholest-3β,5α,6β,19-tetrol and cholestane-3β,5α,6β-triol. As we have mentioned in the Introduction section , 24-methylene-cholest-3β,5α,6β,19-tetrol [2] and cholestane-3β,5α,6β-triol [3, 4] were firstly reported and characterized as cytotoxic natural marine steroids to cancer cells and vascular smooth muscle cells, etc. While our researches have investigated their neuroprotective effects against excitotoxicity, ischemic stroke and spinal cord injury [5, 6]. In order to overcome the cytotoxic effects of these natural marine steroids on normal cells such as vascular smooth muscle cells [7], its derivative of 5α-androst-3β,5α,6β-triol (TRIOL) lacking isooctyl side chain was designed and synthesized [1]. Our results in this MS and previous studies confirmed the potent neuroprotective effect of TRIOL with the structure of polyhydroxysterol [1]. The safety of TRIOL with the structure lacking isooctyl side chain has been preliminarily evidenced by the phase I clinical trial (No. CTR20160737) for acute ischemic stroke in China (data not shown here). 

2.      Indeed, TRIOL has not been reported to be detected or isolated from marine organisms. However, TRIOL is a polyhydroxysterol derivative of natural marine steroids 24-methylene-cholest-3β,5α,6β,19-tetrol and cholestane-3β,5α,6β-triol. 24-methylene-cholest-3β,5α,6β,19-tetrol is a natural tetrahydroxy sterol extracted from soft coral Nephthea brassica collected in South China Sea[2], while cholestane-3β,5α,6β-triol is not only an endogenous metabolite of cholesterol but also a natural marine steroid existed in various marine organisms including soft coral Sinularia sp [3]and gorgonian Menella kanisa[8]. Moreover, we have designed and synthesized their derivative of TRIOL from a marine steroid hormone dehydroepiandrosterone (DHEA) with 3 steps as shown in Scheme 1 below [1]. In summary, TRIOL is a neuroactive derivative of natural marine steroids with similar polyhydroxysterol structure and simple synthetic processes.

Scheme 1. The synthesis of 5α-androst-3β,5α,6β-triol (TRIOL) from DHEA.

(a) KOH, diethylene glycol, hydrazine hydrate, 210 °C. (b) H2O2, HCOOH, RT. (c) NaOH, MeOH, 60 °C. The synthetic routes of TRIOL were previously published in Steroids.78 (2013) 996–1002.

3.      The major purpose of this MS is to evaluate the neuroprotective effects of TRIOL on retina injury in acute glaucoma. As a derivative of marine-derived natural steroids, the pharmacological characterization of TRIOL on its neuroprotective effects and the preliminary investigation of mechanisms on acute glaucoma could be suitable for the scope of Marine Drugs. And we hope this study can be of interest for the readers of Marine Drugs. Undoubtedly, more works still need to be carried out to investigate the structure-function relationships and to optimize the synthetic routes of TRIOL.

4.      At last, the introductions of TRIOL were revised in the updated MS according to your insight comments. Part of the descriptions in Abstract and Introduction were re-composed in Line 26-29 and Line 61-80 respectively.

Point 2: Although generally well written, English language editing is mandatory, particularly the sections of Introduction and Discussion. Examples include:

Line 50: “…followed with the by apoptosis of RGCs…”.

Line 55: Correct “moitics” to “miotics”.

Lines 61-62: “…natural product extracted from the soft coral Nephthea brassica collected in South China Sea…”.

Lines 68-70: “In order to overcome possible side effects of cholestane-3β,5α,6β-

triol on endothelial cell calcification, its the analogue of 5α-androst-3β,5α, 6β-

triol (TRIOL) without lacking the isooctyl side chain were was synthesized.”

Response 2: Thank you for your specific advices on our English writing and we have made correction as follows:

1.      The statement “…followed with the apoptosis of RGCs…” was revised as “…followed by apoptosis of RGCs…” (Line 52);

2.      The statement “moitics” were corrected as “miotics” (Line57);

3.      The statement “…natural product extracted from soft coral Nephthea brassica in South China Sea…” were corrected as “…a tetrahydroxy sterol extracted from soft coral Nephthea brassica collected in South China Sea…” (Line 65-66);

4.      The statement “In order to overcome possible side effects of cholestane-3β,5α,6β-

triol on endothelial cell calcification, its the analogue of 5α-androst-3β,5α, 6β-triol (TRIOL) without the isooctyl side chain were synthesized.” were revised as “In order to overcome possible side effect of cholestane-3β,5α,6β-triol on vascular smooth muscle cell calcification and the cytotoxic potential of 24-methylene-cholest-3β,5α,6β,19-tetrol, their derivative of 5α-androst-3β,5α,6β-triol (TRIOL) lacking isooctyl side chain were designed and synthesized” (Line 76-80).

Point 3: Concerning the experimental data, the authors are requested to clarify the concentrations (80 μg ?) used on the in vivo testing in AIH rat models (Lines 103, 104 and 107). In order to validate the results from western blot analysis, i.e. the interference with the expression Nrf2 and HO-1 (section 2.2), it is also mandatory to include the results concerning the quantification of the relative optical density of the bands.

Response 3: We are really sorry for our negligence for the description of the dosage of TRIOL and we thank you for your vital advices on data presentation of WB. Relative statements in the MS were revised as below:

1.      In order to increase the local concentration of TRIOL in retina, we chose intravitreal injection as the administration route. Limited by the volume capacity of rat vitreous cavity, the maximum injection volume of drugs was 8 μL per eye [9]. As the concentration of TRIOL injection is 10 mg/mL, the high dosage of TRIOL was set to 80 μg/eye and the low dosage were set to 40 μg/eye. Relative statements in section 2.1 of the Results were revised in Line 112-118.

2.      In order to validate the results of WB, the optic density quantification of Nrf2 and HO-1 in Figure 2.A were analysed by Image Lab 4.0 software and shown in the right panel of Figure 2.A (section 2.2). Relative statements about the quantification were added in the figure legend of Figure 2.A (Line 169-170).

References:

1.         Chen, J.; Leng, T.; Chen, W.; Yan, M.; Yin, W.; Huang, Y.; Lin, S.; Duan, D.; Lin, J.; Wu, G.; Zhang, J.; Yan, G., A synthetic steroid 5alpha-androst-3beta,5,6beta-triol blocks hypoxia/reoxygenation-induced neuronal injuries via protection of mitochondrial function. Steroids 2013, 78, (10), 996-1002.

2.         Lu, W.; Zeng, L.; Su, J., Synthesis of polyhydroxysterols (IV): synthesis of 24-methylene-cholesta-3beta,5alpha,6beta,19-tetrol, a cytotoxic natural hydroxylated sterol. Steroids 2004, 69, (7), 445-9.

3.         Sun, H.; Liu, F.; Feng, M. R.; Peng, Q.; Liao, X. J.; Liu, T. T.; Zhang, J.; Xu, S. H., Isolation of a new cytotoxic polyhydroxysterol from the South China Sea soft coral Sinularia sp. Natural product research 2016, 30, (24), 2819-2824.

4.         Wang, P.; Tang, H.; Liu, B. S.; Li, T. J.; Sun, P.; Zhu, W.; Luo, Y. P.; Zhang, W., Tumor cell growth inhibitory activity and structure-activity relationship of polyoxygenated steroids from the gorgonian Menella kanisa. Steroids 2013, 78, (9), 951-958.

5.         Hu, H.; Zhou, Y.; Leng, T.; Liu, A.; Wang, Y.; You, X.; Chen, J.; Tang, L.; Chen, W.; Qiu, P.; Yin, W.; Huang, Y.; Zhang, J.; Wang, L.; Sang, H.; Yan, G., The major cholesterol metabolite cholestane-3beta,5alpha,6beta-triol functions as an endogenous neuroprotectant. The Journal of neuroscience : the official journal of the Society for Neuroscience 2014, 34, (34), 11426-38.

6.         CAI Xiang, Q. P.-x., SU Xing-wen, LI Qun, XU Shi-hai, YAN Guang-mei, The Protection Effects of YC-1 on Cultured Rat Cerebellar Granule Neurons Against Apoptosis Induced by Low Potassium. Acad. J. SUMS. 2000, 21, (3), 161-164.

7.         Liu, H.; Yuan, L.; Xu, S.; Wang, K., Endothelial cell and macrophage regulation of vascular smooth muscle cell calcification modulated by cholestane-3beta, 5alpha, 6beta-triol. Cell biology international 2007, 31, (9), 900-7.

8.         Wang, P.; Tang, H.; Liu, B. S.; Li, T. J.; Sun, P.; Zhu, W.; Luo, Y. P.; Zhang, W., Tumor cell growth inhibitory activity and structure-activity relationship of polyoxygenated steroids from the gorgonian Menella kanisa. Steroids 2013, 78, (9), 951-8.

9.         Di Pierdomenico, J.; Garcia-Ayuso, D.; Jimenez-Lopez, M.; Agudo-Barriuso, M.; Vidal-Sanz, M.; Villegas-Perez, M. P., Different Ipsi- and Contralateral Glial Responses to Anti-VEGF and Triamcinolone Intravitreal Injections in Rats. Investigative ophthalmology & visual science 2016, 57, (8), 3533-44.

Reviewer 3 Report

The manuscript submitted by Long-xiang Sheng and coworkers summarises results demonstrating that 5α-androst-3β,5α,6β-triol, a synthetic marine 2 neuroactive steroid, protects retinal ganglion cells 3 from ischemia-reperfusion injury through activating 4 Nrf2 pathway.  Thus, the authors propose a promising marine drug candidate, TRIOL, for the therapeutics of acute glaucoma. 

This work is of high interest, mainly to readers and professionals related to pharma-therapy and eyes diseases. However, some minor issues must be addressed by the authors before its publication. Some comments and suggestions have been embedded through the manuscript in order to help the authors.

Author Response

Points: This work is of high interest, mainly to readers and professionals related to pharma-therapy and eyes diseases. However, some minor issues must be addressed by the authors before its publication. Some comments and suggestions have been embedded through the manuscript in order to help the authors.

Responses: We truly appreciated reviewer for the comments and advices on the title and context of our MS, relative statements were revised in the MS according to your suggestions.

1. The title of MS was revised as “5α-androst-3β,5α,6β-triol, a marine steroid derivative, protects retinal ganglion cells from ischemia-reperfusion injury through activating Nrf2 pathway”.

2. The statement “Reperfusion of retina after ischemia induces the overproduction of ROS (reactive oxygen species), RNS (reactive nitrogen species) and inflammatory responses in RGCs and microglia.” were revised in the 3rd paragraph of Introduction (Line 88).

3. The statement “We chose intravitreal injection as a route of drug administration in order to increase the local drug concentration in retina. Limited by the volume capacity of rat vitreous cavity, the maximum injection volume of TRIOL (10 mg/mL) was 8 μL per eye. After the AIH treatment on right eyes, the right eyes of each group received an intravitreal administration of 40 or 80 μg TRIOL per eye or 20% HP-β-CD as vehicle respectively, while the left eyes served as normal control.” were revised in section 2.1 of the Results (Line 112-118).

4. According to your comments we re-edited Figure 1 and 2 to improve the quality of our presented data as listed below.

(1) In order to validate the results of WB, the optic density quantification of Nrf2 and HO-1 in Figure 2.A were analysed by Image Lab 4.0 software and shown in the right panel of Figure 2.A (section 2.2). Relative statements about the quantification were added in the figure legend of Figure 2.A (Line 169-170).

(2) We marked RGCs, the axons/dendrites of RGCs and different cell layers in retina with conspicuous arrows in re-edited Figure 1.C/1.F and Figure 2.C/2.E. Relevant descriptions about the modifications mentioned above were added in the figure legends respectively.

(3)  Figure 2 were replaced with a picture with a higher resolution.

5. According to your suggestions on Discussion in section 3, we added some statements in (Line 310-317) to discuss the comparison between TRIOL and other reported agents to reinforce the benefits of TRIOL.

6. Details about the animal experiments including ethics statements and number of animals used in this MS were described in section 4.1 of Materials and Methods. The statement “the experimental projects were approved by the Ethics Committee of Zhongshan Ophthalmic Center of Sun Yat-sen University (approval number: No.2018-111).” (Line 330-333) and “60 SD rats, 40 wild type C57BL6J mice and 40 Nrf2-/- knockout mice were used in this research.” (Line 339-340) were revised in section 4.1.

7. The statement “supplemented with 1× protease inhibitor cocktail (Merck Millipore, Cat No.539131, USA)” was added in section 4.6 in Materials and Methods (Line 407-408). The concentrations of each component in protease inhibitor cocktail were 500 μM AEBSF Hydrochloride, 150 nM Aprotinin, 1μM E-64 protease inhibitor, 0.5 mM EDTA disodium, 1 μM Leupeptin Hemisulfate.

8. The formats of references were checked and revised according to author guidelines.

Reviewer 4 Report

This is a well-executed and designed studies on the potential use of TRIOL to improve acute glaucoma. I have several comments that I would like the authors to clarify.

Page 3 Lines 103/104/107: μg not ug

Section 2.1 / Figure 1: I am confused about the results of the data. It appears that the addition of the vehicle HP-β-CD actually lowered the number of RGCs. I presume those experiments with TRIOL also includes HP-β-CD, so the authors should include that in the labelling. Whilst the data suggests TRIOL may improve AIH, the choice of HP-β-CD as vehicle should be revisited because it appears to make AIH worse – perhaps with a better vehicle, the dose of TRIOL that is needed would be less.

Section 2.2 / Figure 2: The authors clearly demonstrated the effect of TRIOL is linked to Nrf2 by using WT and Nrf2 knockout mice.

Section 2.3 / Figure 3: In Figure 3C, HP-β-CD on its own appears to lower the mRNA expression of TNF-α, CXCL-10 and CCL-2 – why is that?

Sections 2.4 / 2.5: The authors clearly showed that TRIOL induces nuclear translocation Nrf2, and are involved in activating the Nrf2/HO-1 pathway, as demonstrated by the upregulates the expression of Nrf2 and HO-1.

Author Response

This is a well-executed and designed study on the potential use of TRIOL to improve acute glaucoma. I have several comments that I would like the authors to clarify.

We truly thank reviewer for the comments and valuable advices on our MS which will greatly improve our future research. The revisions of relative statements were listed below:

Point 1: Page 3 Lines 103/104/107: μg not ug

Response 1: Thank you for your kindly pointing out our mistakes in English writing. All the statements “ug” used as units were corrected as “μg”

Point 2: Section 2.1 / Figure 1: I am confused about the results of the data. It appears that the addition of the vehicle HP-β-CD actually lowered the number of RGCs. I presume those experiments with TRIOL also includes HP-β-CD, so the authors should include that in the labelling. Whilst the data suggests TRIOL may improve AIH, the choice of HP-β-CD as vehicle should be revisited because it appears to make AIH worse – perhaps with a better vehicle, the dose of TRIOL that is needed would be less.

Response 2:

1. As is mentioned in Methods and Materials section, the injection of TRIOL used in this MS was dissolved by vehicle of 20% HP-β-CD, we thought it’s not necessary to specifically label the statement “TRIOL” as “TRIOL+20% HP-β-CD”.

2. Although it seems that the addition of the vehicle HP-β-CD actually lowered the number of RGCs in Figure 1.D, the statistical analysis has shown that there is no significant difference in the number of RGCs between AIH group and AIH + HP-β-CD group, which demonstrated that the treatment of vehicle HP-β-CD has no significant influence on the number of RGCs in the current dosage. The statistic reports of one-way ANOVA and Dunnett’s post hoc test were shown below.

3. Additionally, Hydroxypropyl-β-Cyclodextrin (HP-β-CD) has been proved to be a safe and qualified pharmaceutical vehicle to improve the solubility of drugs such as Itraconazole, Mitomycin, etc [1]. The possible toxicity of HP-β-CD as a vehicle may be introduced by the side product of cyclodextrin during the synthesis processes [1]. However, the vehicle of 20% HP-β-CD shows the best solubilizing ability to TRIOL so far among various kinds of vehicles we have tried. The injection of TRIOL (containing 20% HP-β-CD) and the vehicle have been proved to be safe in phase I clinical trial (No. CTR20160737) for acute ischemic stroke in China (data not shown here). Undoubtedly, more works are still required to investigate the specific pharmaceutical preparations of TRIOL for the clinic use in acute glaucoma.

Point 3: Section 2.3 / Figure 3: In Figure 3C, HP-β-CD on its own appears to lower the mRNA expression of TNF-α, CXCL-10 and CCL-2 – why is that?

Response 3: We truly appreciate your comments on our data consistency here. We admit that there is a big error bar between the LPS and LPS+HP-β-CD group in Figure 3.C although the experiments had been repeated for 3 times with similar tendency. Despite the lower trends of the mRNA expression of TNF-α, CXCL-10 and CCL-2 in LPS + HP-β-CD group, there are no statistically significant differences between LPS and LPS+ HP-β-CD group. We have to say the BV2 microglia cells are easily activated by multiple factors during the in vitro experiments which may contribute to the observations above. However, our results here investigated that TRIOL can significantly inhibit the mRNA expression of pro-inflammatory factors TNF-α, IL-1β, CXCL-10 and CCL-2. To further confirm the inhibition of microglial activation by TRIOL, we not only evaluate the activated morphology of BV2 in vitro (Figure 3.A/B) and in vivo (Figure 3.D/E) but also detect the mRNA expression of inflammatory factors as a functional detection of microglial activation (Figure 3.C). Moreover, consistent with the results in Figure 3.C, the results of activated microglial numbers in Figure 3.A/B and 3.D/E further confirmed that TRIOL, but not vehicle of HP-β-CD, can inhibit the inflammatory activation of microglia.

1. Brewster, M. E.; Loftsson, T., Cyclodextrins as pharmaccutical solubilizers. Adv. Drug Deliv. Rev. 2007, 59, (7), 645-666.

Round  2

Reviewer 2 Report

While the current version of the manuscript marinedrugs-489907 was significantly improved by the authors, minor issues concerning English language style and editing should be considered before its final acceptance. In fact, the authors are advised to have the manuscript revised by an English native speaker or an editing service.

Examples can be found throughout the whole manuscript as in:

Line 28: “…isolated from the soft coral Nephthea brassica and the gorgonian Menella kanisa, respectively.”

Line 63: Please revise “24-methylene-choles-…” to “24-Methylene-choles…”.

Line 64: “…extracted from the soft coral…”.

Lines 66/67: “…was also reported with to exhibit cytotoxic activities to towards some cancer cells.”.

Lines 81/82: Consider “Our previous in vitro studies have identified demonstrated that TRIOL can serve as a potent neuroprotectant in vitro via protecting…”.

Line 126: “…TRIOL can dose-dependently attenuated the injury and keep, without noticeable interference on the integrity of axon networks.”

Lines 150/151: Revise “wide-type” to “wild-type” and “didn’t” to “did not”.

Authors are also advised to consider the Instructions for Authors as several references (for example references 3-8) were described in disagreement with it.

While I continue to consider that the naturally-occurring compounds that inspired the synthesis of TRIOL have a ubiquitous distribution, the current study was correctly designed, the obtained results being relevant. As such, if the editors consider that the content is aligned with the scope of Marine Drugs, I consider that the current manuscript should be considered for publication after English language editing.

Author Response

Dear reviewer,

We thank you very much for your time and providing valuable suggestions on our manuscript entitled “Marine steroid derivative 5α-androst-3β, 5α, 6β-triol protects retinal ganglion cells from ischemia-reperfusion injury through activating Nrf2 pathway” (ID: marinedrugs-489907). We have revised the manuscript carefully according to your comments, which are very helpful for improving our manuscript. Furthermore, according to your suggestions, we have our manuscript revised extensively by the MDPI English editing service, with a certificate as a supplemented file. All revisions have been clearly highlighted using the "Track Changes" function. We wish that our revised manuscript will meet with approval. The main revisions in the manuscript are marked with red and listed below:

Point 1: While the current version of the manuscript marinedrugs-489907 was significantly improved by the authors, minor issues concerning English language style and editing should be considered before its final acceptance. In fact, the authors are advised to have the manuscript revised by an English native speaker or an editing service.

Response 1: Thank you for your kindly pointing out our mistakes in English writing. According to your suggestions, related statements were revised as below:

1)        Line 28-29: “isolated from soft coral Nephthea brassica and gorgonian Menella kanisa, respectively.” was revised as “isolated from the soft coral Nephthea brassica and the gorgonian Menella kanisa, respectively.”;

2)        Line 64: “24-methylene-choles-…” was revised as “24-Methylene-choles…”;

3)        Line 65: “extracted from soft coral…” was revised as “extracted from the soft coral…”;

4)        Line 67-68: “was also reported with cytotoxic activities to some cancer cells.” was revised as “was also reported to exhibit cytotoxic activities towards some cancer cells.”;

5)        Lines 81-82: “Our previous studies have identified that TRIOL can serve as a potent neuroprotectant in vitro via protecting…” was revised as “Our previous in vitro studies have demonstrated that TRIOL can serve as a potent neuroprotectant by protecting…”;

6)        Line 125-126: “TRIOL can dose-dependently attenuated the injury and keep the integrity of axon networks.” was revised as “TRIOL dose-dependently attenuated the injury and kept the integrity of axon networks.”;

7)        Line 150-152: “wide-type” was revised as “wild-type”, and “didn’t” was revised as “did not”.

 Moreover, according to your suggestions, the English language and formats of this manuscript were edited extensively by the MDPI English editing service. Relative revisions were all clearly highlighted in the updated manuscript using the “Track Changes” function. According to the suggestions of MDPI English editors, the title of this manuscript was revised as “Marine steroid derivative 5α-androst-3β, 5α, 6β-triol protects retinal ganglion cells from ischemia-reperfusion injury through activating Nrf2 pathway”.

Point 2: Authors are also advised to consider the Instructions for Authors as several references (for example references 3-8) were described in disagreement with it.

Response 2: In the References section, the statement “Investigative ophthalmology & visual science” was revised to “Investigative ophthalmology and visual science” in the journal name of references 4, 38, 39, 40, 42. The descriptions of relative references were checked and new references 12, 14, 15, 18 were added.

Point 3: While I continue to consider that the naturally-occurring compounds that inspired the synthesis of TRIOL have a ubiquitous distribution, the current study was correctly designed, the obtained results being relevant. As such, if the editors consider that the content is aligned with the scope of Marine Drugs, I consider that the current manuscript should be considered for publication after English language editing.

Response 3: Thank you for the kind comments and concerns about this study. We report in this manuscript that TRIOL, a marine steroid derivative, can protect retina from ischemia-reperfusion injury, which suggests TRIOL as a promising candidate drug for acute glaucoma. We believe that, in the near future, more and more effective neuroactive steroids with novel structures can be discovered to treat acute glaucoma based on TRIOL.
